# Vulnerability Assessment of Pelagic Sharks in the Western North Pacific by Using an Integrated Ecological Risk Assessment

**DOI:** 10.3390/ani11082161

**Published:** 2021-07-21

**Authors:** Kwang-Ming Liu, Lung-Hsin Huang, Kuan-Yu Su, Shoou-Jeng Joung

**Affiliations:** 1Institute of Marine Affairs and Resource Management, National Taiwan Ocean University, Keelung 20224, Taiwan; resumption2@gmail.com (L.-H.H.); supipi76@gmail.com (K.-Y.S.); 2George Chen Shark Research Center, National Taiwan Ocean University, Keelung 20224, Taiwan; f0010@mail.ntou.edu.tw; 3Center of Excellence for the Oceans, National Taiwan Ocean University, Keelung 20224, Taiwan; 4Department of Environmental Biology and Fisheries Science, National Taiwan Ocean University, Keelung 20224, Taiwan

**Keywords:** demographic analysis, productivity, susceptibility, intrinsic rate of population growth, fishing impact

## Abstract

**Simple Summary:**

A new integrated ecological risk assessment (ERA) including the IUCN Red List category, the body weight variation trend of 1989–2011 with large sample size (*n* > 678,000), and the inflection point of population growth curve coupled with the ERA was developed to assess the impact of longline fishery on the pelagic sharks in the western North Pacific. The intrinsic rate of population growth was used to estimate the productivity, and the susceptibility was estimated by the multiplication of the catchability, selectivity, and post-capture mortality. Five groups were identified based on the cluster analysis coupling with non-parametric multi-dimensional scaling. Rigorous management measures are recommended for the scalloped hammerhead, silky, and spinner shark at highest risk, setting total allowable catch quota is recommended for the bigeye thresher, and sandbar shark, and a consistent monitoring scheme is suggested for the smooth hammerhead, shortfin mako, pelagic thresher, oceanic whitetip, and dusky shark.

**Abstract:**

The vulnerability of 11 pelagic shark species caught by the Taiwanese coastal and offshore longline fisheries in the western North Pacific were assessed by an ecological risk assessment (ERA) and 10 of the 11 species was assessed by using an integrated ERA developed in this study. The intrinsic rate of population growth was used to estimate the productivity of sharks, and the susceptibility of sharks was estimated by the multiplication of the catchability, selectivity, and post-capture mortality. Three indices namely, the IUCN Red List category, the body weight variation trend, and the inflection point of population growth curve coupled with ERA were used to conduct an integrated ERA. The results indicated that the scalloped hammerhead is at the highest risk (group 1), followed by the silky shark, and the spinner shark at high risk (group 2). The bigeye thresher, and sandbar shark fall in group 3, the smooth hammerhead falls in group 4, and the shortfin mako, pelagic thresher, oceanic whitetip, and dusky shark fall in group 5. Rigorous management measures for the species in groups 1 and 2, setting total allowable catch quota for group 3, and consistent monitoring schemes for groups 4 and 5 are recommended.

## 1. Introduction

Most pelagic sharks are apex predators in the ocean, they can maintain the balance of marine ecosystem and thus play an important role in the ecosystem [1,2,3]. Several studies have indicated the trend of pelagic fish species are shrinking [4,5,6,7,8] and the latest study indicated that global abundance of 74% of 31 pelagic shark and ray species have declined by 71% since 1970 [9]. Increasing fishing efforts and expanding fishing grounds, not only led to the reduction of target species, habitat destruction, and decreased the size at catch, but also resulted in dramatic changes in marine ecosystems directly or indirectly [10,11,12]. In addition, the marine ecosystem structure may be altered when constantly removing the by-catch species such as sharks, rays, and marine mammals [13]. In view of this, how to effectively manage marine resources has become an important issue for tuna regional fisheries management organizations (tRFMOs).

In the past, most fish stock assessment studies focused on single species assessment. However, for stocks with poor data or limited biological information, risk assessment methods were usually used by applying a qualitative or semi-quantitative approach. In recent years, the ecological risk assessment (ERA) method such as productivity-susceptibility analysis (PSA) has been proposed for fisheries management, marine animals can be analyzed based on existing biological and fishery information [14]. It is a useful methodology for assisting fisheries management from an ecosystem perspective [15]. The method is calculated by combining the productivity of animals and the susceptibility to fisheries to assess the stock’s relative vulnerability [16]. Based on the results from the ERA, the relative risk of animals, and the prioritization of management and conservation can be identified.

The ERA method has been proposed to use by various tRFMOs to evaluate the vulnerability of animals, including the species caught by tuna fisheries in the Western and Central Pacific Fishery Commission (WCPFC) waters [17], sharks caught by various fisheries in the Indian Ocean Tuna Commission (IOTC) waters [18,19], and sharks in the International Commission for the Conservation of Atlantic Tunas (ICCAT) waters [20,21] as well as national organization such as the Australian Fisheries Management Authority (AFMA) [22]. Furthermore, the ERA methods have been used to assess the vulnerability of target and bycatch species stocks to fishing gear, including the Atlantic tuna fisheries by the United States and European Union [15], Alaska demersal fishery [16], and Australian trawl fishery [23]. In addition, Stelzenmüller et al. [24] proposed the spatial risk assessment framework for the UK continental shelf fish and shellfish. Chin et al. [25] analyzed the vulnerability of sharks and rays in Australia’s Great Barrier Reef by taking account the climate change effect. Gallagher et al. [26] reviewed the ERA and its application to elasmobranch conservation and management.

Although ERA has been used by WCPFC on assessing the risk of species caught in the western and central Pacific Ocean (WCPO) tuna fisheries, such an approach has never been applied to the multi-species management in the western North Pacific Ocean despite a recent study by Lin et al. [27]. The authors used a semi-quantitative PSA on 52 species including three shark species—the silky, *Carcharhinus falciformis*, blue shark, *Prionace glauca*, and shortfin mako shark, *Isurus oxyrichus* in five fisheries in eastern Taiwanese waters. However, other pelagic shark species commonly caught by the Taiwanese coastal and offshore longline fishing vessels were not included in their study. Annual yield of pelagic sharks (excluding blue shark), caught in the western North Pacific, landed at Nanfangao fishing port, northeastern Taiwan increased from 1724 tons in 1989 to the peak of 2876 tons in 1996 and decreased thereafter to around 1500 tons after 2005 based on the sales records. Species composition in terms of weight indicated that the blue shark was the dominant species, followed by the shortfin mako and the scalloped hammerhead. Although stock assessments of some of these species such as pelagic thresher [28,29], bigeye thresher [30,31], shortfin mako [32,33,34], and smooth hammerhead [35] in the region have been conducted using demographic or per recruit models. The relative risk on the pelagic sharks from fisheries in the region has not been evaluated, thus, this study attempts to assess the vulnerabilities of 11 pelagic shark species commonly caught by the Taiwanese coastal and offshore longline fisheries in the western North Pacific.

In addition to conventional PSA, an integrated ERA combining the ERA with species’ endangered status and the inflection point of population growth curve has been developed to better describe the ecological risk of sharks and rays [20]. However, the size variation of these elasmobranchs has not been taken into account due to unavailability of the data. In this study, as the fishing ground and fishing technique of the Taiwanese longline fishing vessels in the western North Pacific did not change, the long-term body weight variation trend (1989–2011) of pelagic sharks can better represent the fishing impact on species population. Hence, an integrated ERA combining the ERA, IUCN Red List index, annual body weight variation trend, and the inflection point of population growth curve was developed to assess the vulnerability of pelagic sharks in the western North Pacific. It is hoped that the results derived from the present study can provide useful information for prioritizing management measures to achieve the goal of sustainable use of shark resources in this region.

## 2. Materials and Methods

### 2.1. Study Area and Species

This study covered the waters in the western North Pacific ranging from 20° N to 30° N, and from 120° E to 140° E where was the conventional fishing ground of the Taiwanese coastal and offshore longline fishing vessels (<100 gross tonnage) (Figure 1). Most of these longline fishing vessels mainly target dolphinfish *Coryphaena hippurus*, tunas, and billfishes (sharks are the bycatch) from April to October and switch to targeting sharks (seasonal shark targeting) by changing gear configuration from November to next March. Eleven pelagic shark species, commonly caught by these fishing vessels, namely, the bigeye thresher *Alopias superciliosus*, pelagic thresher, *A. pelagicus*, silky shark, spinner shark, *C. brevippina*, dusky shark, *C. obscurus*, oceanic whitetip, *C. longimanus*, sandbar shark, *C. plumbeus*, shortfin mako shark, blue shark, scalloped hammerhead, *Sphyrna lewini*, and smooth hammerhead, *S. zygaena* were analyzed in this study.

### 2.2. Source of Data

Species-specific landing data, including the whole (body) weight of each individual fish (except very few small-size individuals that were weighed in group and only average weights were available) at the Nanfangao fishing port, northeastern Taiwan from 1989–2011, were collected from the sales records. These data were used to estimate the species-specific catch in number, the species composition, and the mean and median weight. The individual body weight data were converted to length via an existing length–weight relationship in this region. The age of each individual was then estimated by substituting the converted length into the growth equation of each species (Table 1).

### 2.3. Demographic Analysis

Demographic parameters of each species were estimated using Krebs [48] equations assuming sex ratio was 1:1 for the embryos and the population was in equilibrium condition (∑$\frac{1}{2}m_{x}l_{x}e^{-rx}$ = 1) based on existed life history parameters (Table 1 and Table 2). The demographic parameters were calculated as follows:(1)$$R_{0}=\sum_{x=0}^{t_{max}}\frac{1}{2}m_{x}l_{x},\;G=\frac{\sum_{x=0}^{t_{max}}\frac{1}{2}xl_{x}m_{x}}{R_{0}},\;r=\frac{\ln R_{0}}{G},\;\lambda =e^{r}$$
where *R*_0_: the net reproductive value per generation, *x*: age, *t_max_*: the maximum age, *m_x_*: the fecundity of age *x*, *l_x_*: survival rate until age *x*, G: generation time, r: intrinsic rate of population growth, λ: finite rate of population increase, t_x2_: theoretical population doubling or halving time.

### 2.4. Ecological Risk Assessment (ERA)

ERA considers productivity and susceptibility of sharks. Productivity is the ability of withstanding the exploitation by replacing individuals through the reproduction, survival, and growth of individuals. Usually, the intrinsic rate of population growth (r) is expressed as the productivity. Susceptibility is the impact of a fishery on a species population and is estimated by the multiplication of the probabilities of the following three parameters: (1) catchability: the species composition (percentage of catch in weight) of 11 shark species based on the landing data from 1989–2011; (2) selectivity, the ratio of age range of catch (the maximum age at catch–minimum age at catch) and the longevity [22]; and (3) post-capture mortality, including the mortality of retention and those after being discarded or live released. Based on the fisherman interview, despite the releasing of oceanic whitetip and silky shark which were non-retention species in the WCPO after 2013 and 2014, only very small portion of other shark catch was discarded or released. Thus, the post-capture mortality was referred to Cortés et al. [54] that estimated from the observed data of US far sea fishery.

Susceptibility was estimated from the multiplication of the aforementioned three parameters. When the productivity and susceptibility were estimated, a modified Euclidean distance (D) [20] was used to estimate the vulnerability as:(2)$$D\;=\;\sqrt{{\left(p-0.5\right)}^{2}+{\left(s-0\right)}^{2}}$$
where *p* is productivity, s is susceptibility. Index of body weight variation trend (*S_w_*).

IUCN Red list index (C)

The endangered status of the 11 pelagic shark species was evaluated based on the latest IUCN Red List assessment of pelagic sharks [55]. To quantify the risk of Red List species, we followed the definition proposed by Simpfendorfer et al. [20], C ranges from 0 and 1 based on the endanger condition. The C = 1 for critical endangered (CR) species, C = 0.8 for endangered (EN) species, C = 0.6 for vulnerable (VU) species, C = 0.4 for near threatened (NT) species, and C = 0.2 for species of least concern (LC) (Table 3).

Body weight variation trend (*S_w_*)

Annual median weight of each shark species was calculated from the landing data of Nanfangao fish market from 1989 to 2011. The slope of the simple linear regression between annual median weight and year was used as the index of body weight variation trend (*S_w_*). The negative value of slope indicated a decrease of size at catch. The value of *S_wa_* ranged from 0–1 by taking an absolute value of *S_w_*.

Inflection point of population growth curve (I)

The inflection point of population growth curve (I) is the ratio of the biomass at maximum sustainable yield (B_MSY_) and the initial biomass (B_0_). The larger the value is, the higher the exploitation is and the less biomass can be utilized. The I value can be estimated as: I = 0.633–0.187(ln(rG)) [56], where G is the generation time in years.

### 2.5. Integrated Ecological Risk Assessment

An integrated ERA was developed based on the combined information of the ERA, C, I, and *S_w_*. As most blue sharks were processed on the sea, the individual weight data only available for very small portion of the blue shark landings. Although the sales data of frozen meat, internal organs, and fins were available after 2001, the catch in number data were still lacking which hindered the estimation of individual weight of blue sharks. Due to lack of *S_w_* information, this species was not included in the integrated ERA assessment (Figure 2). Multivariate analyses including principal component analysis (PCA), and cluster analysis (CA) and non-parametric multi-dimensional scaling (NMDS) have been used in grouping of sharks, skates, and rays based on their life history parameters and habitat information [57,58]. The CA coupled with NMDS was used to group the 10 species based on their similarity of the four indices (ERA, C, *S_wa_*, and I) to conduct the integrated ERA using Primer V. 6 [59].

## 3. Results

### 3.1. Demographic Parameters

Estimated demographic parameters of 11 species are in Table 4. The blue shark has the highest intrinsic rate of population growth (r = 0.351 year^−^^1^), followed by the scalloped hammerhead (r = 0.243 year^−^^1^), while the bigeye thresher has the lowest value (r = 0.008 year^−^^1^), and other species range from 0.016–0.154 year^−^^1^. Without fishing mortality, λ ranged from 1.008 for the bigeye thresher to 1.420 for the blue shark with only two species (the blue shark and scalloped hammerhead) were greater than 1.20. The values of λ for the silky, smooth hammerhead, dusky, and oceanic whitetip shark ranged from 1.088 to 1.166, while λ was smaller than 1.07 for the pelagic thresher, bigeye thresher, spinner, sandbar, and shortfin mako sharks (Table 4).

### 3.2. Ecological Risk Assessment (ERA)

The values of susceptibility ranged from 0.0073 (oceanic whitetip) to 0.2387 (blue shark) and can be categorized as three groups: (1) susceptibility = 0.2387 (blue shark), (2) susceptibility = 0.1069–0.1754 (shortfin mako, scalloped hammerhead), (3) susceptibility = 0.0073–0.0645 (pelagic thresher, bigeye thresher, spinner, oceanic whitetip, dusky, sandbar, silky, and smooth hammerhead shark) (Table 5).

The risk based on Euclidean distance (D) ranged from the highest (D = 0.5017) for the shortfin mako to the lowest (D = 0.2784) for the scalloped hammerhead (Table 6). The higher risk of the shortfin mako, bigeye thresher, sandbar, pelagic thresher, and spinner sharks is mainly due to their low productivity despite various susceptibility. On the other hand, the silky, and oceanic whitetip sharks have lower risk because of their higher productivity and lower susceptibility (Table 6; Figure 3).

### 3.3. IUCN Red List Index (C)

Based on the latest IUCN red list assessment of pelagic sharks [55], the blue, and spinner shark fall in the near threshed (NT, C = 0.4), bigeye thresher, sandbar, silky sharks, and smooth hammerhead fall in vulnerable (VN, C = 0.6), pelagic thresher, shortfin mako, and dusky shark fall in endanger (EN, C = 0.8), while the scalloped hammerhead and oceanic whitetip shark fall in critical endangered (CR, C = 1.0), which have the highest extinction risk (Table 7).

### 3.4. Index of Body Weight Variation Trend (S_w_)

The variations of annual median weight for each species estimated from over 678,000 individual weight records indicated that the scalloped hammerhead shark had the largest decline (*S_w_* = −1.8279) (Table 7), dropping from the peak of 63.4 kg in 1989 to less than 21.2 kg after 2010 (Appendix A). The silky shark had the second largest decline in median weight (*S_w_* = −1.2762) (Table 7), dropping from 40–50 kg before 1997 to less than 30 kg after 2010 (Appendix A). The spinner shark had large variations in annual median weight, while the oceanic whitetip and pelagic thresher shark did not have remark changes in their median weights during the period of 1989–2011 (Appendix A).

### 3.5. Inflection Point of the Population Growth Curve (I)

The estimated I for 11 shark species ranged from 0.4286 for the blue shark to 0.9870 for the bigeye thresher shark (Table 7).

### 3.6. Integrated Ecological Risk Assessment

Five groups were categorized based on the ERA, C, *S_wa_*, and I using the cluster analysis: group (1) the highest risk including the scalloped hammerhead shark, group (2) high risk including the silky, and spinner shark, group (3) median risk including the bigeye thresher, and sandbar shark, group (4) less risk including the smooth hammerhead, and group (5) least risk including the oceanic whitetip, pelagic thresher, shortfin mako, and dusky shark (Figure 4). Similar results were obtained from the NMDS indicating five groups of ecological risk (Figure 5) even the weighting of *S_wa_* was set as 0.5.

## 4. Discussion

This study provides the first assessment of ERA for the 11 shark species (integrated ERA for 10 shark species) in the western North Pacific. The results derived from the present study can be used as reference for setting the priority of conservation and management measures of these species in this region.

### 4.1. Assumptions in This Study

The historical species composition of shark catch, sex ratio, length–weight relationship, age-structure, and proportion of maturity were assumed to be the same as those used in this study. The results derived from this study may be affected if these assumptions are violated. Two stocks—Northern and southern populations of blue sharks in the Pacific Ocean were identified based on tag–recapture data [60]. A single stock of shortfin mako shark was assumed in the North Pacific Ocean based on evidence from genetics, and tagging studies but catch and biological data indicated regional sub-stocks may exist [61]. Although two management stocks (east and west) of the silky shark in the Pacific Ocean were suggested based on genetic analysis [62], one Pacific-wide stock was assumed in the stock assessments of the silky shark [63] and the oceanic whitetip shark [64] without solid evidence. Apart from the blue and shortfin mako shark, the assumption of unit stock for the remaining 9 species in the present study should be further validated using tag-recapture study or molecular techniques.

### 4.2. Analysis of Landing Data

As the body weight variation trend was not available, the blue shark was excluded in the integrated ERA analysis. However, we believed that the blue shark was still at the least risk among all shark species if it was included in the integrated ERA analysis (Table 7).

Besides the Taiwanese fishing vessels, some Japanese longline fishing vessels also operate in the study area. As the fisheries information including species-specific catch in number and weight of these vessels were not available, their impact on sharks could not be assessed. The results of the present study can be improved if the aforementioned data can be incorporated in future analysis.

### 4.3. Analysis of Life History Parameters

The uncertainty of life history parameter estimations was often due to small sample size as the collection of shark sample was difficult. The life history parameters used in this study were adopted from the literature with considerable sample size collected in the western North Pacific. Therefore, we believe these values can represent the life history characteristics of these 11 species. Age and growth parameters of 11 species used in this study was based on vertebral band counting but only bigeye and pelagic threshers were verified with length-frequency analyses. Campana [65] pointed out that bias of age estimation may occur using hard part as ageing character. In addition, the von Bertalanffy growth function may not necessarily provide the best fit for all shark species [66]. Therefore, a multi-model approach with larger sample size and various ageing techniques should be considered in future age and growth study for sharks [67]. In addition, there was uncertainty on vertebral band pair deposition period for the scalloped hammerhead and shortfin mako shark. We adapted the biannual deposition for scalloped hammerhead from Chen et al. [51] and this was supported by Anislado-Tolentino and Robinson-Mendoza [68] in Mexican waters. However, other authors reported annual deposition in other regions of the Pacific Ocean [69,70,71]. As for the shortfin mako shark, annual deposition used in this study was adopted from Hsu [72] and Semba et al. [73] in Northwest Pacific. Ribot-Carballal et al. [74] also concluded annual deposition of growth band pair but Wells et al. [75] concluded a biannual cycle of vertebral band-pair deposition up to 5 years old, whereas Kinney et al. [76] concluded an annual cycle for those older than 5 years old in the eastern North Pacific Ocean. If aforementioned uncertainties in life history parameters were taken account in the integrated ERA as different scenarios, the scalloped hammerhead was still in the highest ecological risk group and the shortfin mako shark fell in the least risk group suggesting that our results were robust.

The intrinsic rate of population growth (r), estimated from demographic analysis, was used as productivity in this study. However, uncertainties of life history parameters and natural mortality that may lead to a biased estimation of r [77] by using this approach. Therefore, stochastic approach should be considered to obtain a more robust estimate of intrinsic population growth rate in the future.

### 4.4. Ecological Risk Assessment

ERA indicated that the shortfin mako, bigeye thresher, sandbar, and pelagic thresher shark were at the highest ecological risk. Similar findings have been reported by Simpfendorfer et al. [20] and Cortés et al. [54]. These authors found that the silky, shortfin mako, and bigeye thresher shark had the highest risk although the dusky and sandbar sharks were not included in their analysis. The high risk of these four species is very likely due to their low productivity. Particularly, the late age at maturity and three-year reproductive cycle yielded the second lowest productivity for the shortfin mako shark. This low productivity coupled with high susceptivity make it rank the highest risk species. Comparing with other teleost species, Lin et al. [27] concluded that sharks (silky, blue, and shortfin mako shark) have the highest vulnerability among 52 fish species caught in eastern Taiwan waters based on a semi-quantitative ERA.

The present study used ERA to assess the risk of exploitation for the 11 shark species in the western North Pacific but some these sharks are highly migratory species such as blue shark, shortfin mako, and oceanic whitetip that widely distribute in the ocean and they may migrate beyond the study area. However, due to lacking of specific fishing location by set, the fishing ground was based on the records by the sampling vessels and fisherman interview. Therefore, the availability—overlapping between the geographic distribution of sharks and the longline fishing ground assumed to be the same for all species was not used in this study. The information of vertical movement range for the 11 shark species in the western North Pacific is still little known despite Musyl et al.’s [78] description of the bigeye thresher, blue, oceanic whitetip, silky, and shortfin mako in the eastern/central Pacific by using tagging experiment. Due to the uncertainty, the availability and encounterability was replaced by the catchability estimated based on the mean percentage of shark catch in weight from 1989 to 2011. We believe this long-term historical landing data of large sample size (*n* > 678,000) can better described the species-specific vulnerability to longline fishery. Future study should focus on the tagging research to understand the vertical and horizontal distribution and movement of the sharks to improve the parameter estimation of availability and encounterability.

The selectivity was estimated based on the ratio of age range of catch and the longevity. The age range was estimated by converting individual shark landing data (body weight) to age which were then used to estimate the minimum and maximum ages of each species. We believed that our estimates based on long-term data of large sample size were representative. As individual body weight data were not available for most blue sharks, the selectivity of blue shark being set as 1 was assumed due to its wide range of size at catch. We believe this was a reasonable assumption.

The post capture mortality used in this study was adopted from Cortés et al. [54] based on the US observer’s data on Atlantic sharks except the oceanic whitetip and silky shark. However, these values may not be representative for the sharks in the western North Pacific. The post release mortality of several pelagic shark species using satellite popup tagging techniques has been documented in recent years. Musyl et al. [78] reported the post release mortality and vertical movement of five pelagic shark species in the central North Pacific. Musyl and Gilman [79] and Schaefer et al. [80,81] demonstrated the post release mortality of blue shark and silky shark in the Pacific Ocean. Campana et al. [82] and Santos et al. [83] also documented the information of shortfin mako in the Atlantic Ocean. However, these updated estimations of post release mortality were much smaller than those reported by Simpfendorfer [20] and Cortés et al. [54] based on the onboard observer’s records. The possible reason was due to different definitions among studies. The post capture mortality defined in this study include retention and mortality of live release (post release mortality). It is likely that the tagging experiments were based on live (healthy or minor injury) individuals but the observer’s estimation was from all individuals landed on deck. To improve our results, further study on the post release mortality for other species should be conducted in the future.

### 4.5. IUCN Red List Index

The IUCN red list assessment of pelagic sharks was based on life history parameters, abundance trend, fishery data, and expert opinions. Dulvy et al. [84] suggested that the assessment result derived by the IUCN Red List is a good index that can represent the stock status of sharks when the full stock assessment is lacking. In the present study, the endanger index was estimated based on the latest global pelagic shark assessment results [55] and was believed to be representative. However, this information was in global scale but was not specifically for the western North Pacific. If regional (Indo-west Pacific) assessment results are available in the future, that information should be used in an updated analysis.

### 4.6. Integrated ERA

Simpfendorfer et al. [20] suggested that the scalloped hammerhead falls in the group of low risk in the Atlantic Ocean. The authors reported that selectivity being 0.11, susceptibility being 0.06, IUCN index of 0.4 (near threatened) [55] with ERA of 0.42 for scalloped hammerhead. In the present study, we estimated the selectivity to be 0.968, susceptibility of 0.1069, ERA of 0.2784 with IUCN index of 1.0 [55] combined with the largest decline of median weight among 10 species concluding this species has the highest risk. The recent promoting of IUCN red list from EN to CR for scalloped hammerhead in global scale [85] is another reason that resulted in different results in these two studies. We believed our integrated ERA assessment of scalloped hammerhead can better represent the stock status of this species in the western North Pacific.

The decline of median weight from 43 kg in 1989 to 33 kg in 2010 for the silky shark suggesting over-exploitation of this species is likely the reason that this species had the second highest risk among the 10 shark species. Cortés et al. [54] conducted the ERA on the pelagic sharks and reported that the silky shark had the highest risk which was comparable with the result derived from this study. The management measure of ban retention for this species has been taken in WCPFC and ICCAT.

Simpfendorfer et al. [20] suggested that the shortfin mako and bigeye thresher had the highest risk among the pelagic sharks in the Atlantic Ocean. However, we concluded the shortfin mako and bigeye thresher in the least risk group in the western North Pacific. The possible reason is the body weight of these two species in the western North Pacific did not have significant decline (*S_w_* = −0.436 and −0.282) compared with the scalloped hammerhead and silky shark and the body weight variation was not considered by Simpfendorfer et al. [20]. In addition, different stock status of the shortfin mako shark in the two oceans was likely another reason. The latest North Pacific shortfin mako shark stock assessment indicated that there was no overfished and overfishing was not occurring [59] suggesting the assessment of shortfin mako derived from integrated ERA in this study was reasonable. However, the recent stock assessment indicated the North Atlantic shortfin mako was overfished and overfishing was continuing, similar situation may occur in South Atlantic stock [86]. In summary, the inclusion of the index of body weight variation trend in the integrated ERA used in this study can provide better assessment on the risk of over-exploitation for sharks in this region.

### 4.7. Uncertainty

Accurate stock assessment is difficult as it is hard to collect accurate biological and fisheries information. Therefore, management regulations based on various management schemes may be set by fishery managers [87]. If the uncertainty was resulted from artificial factors, this can be reduced by collecting more accurate data [88]. Therefore, to reduce the uncertainty, the biological and fisheries data should be updated to improve the accuracy of ERA and integrated ERA.

### 4.8. Current Management Measures

Of the 11 species analyzed in this study, the oceanic whitetip shark and silky shark have been banned for commercial retention in the WCPO by WCPFC since 2013 and 2014, respectively. No specific management measures have been taken for other shark species in this region besides the catch reporting scheme. Based on the vital parameter analysis of 38 species of sharks, Liu et al. [57] suggested that protection of adults or TAC management measure should be taken for shark species of slow growing and small litter size such as bigeye thresher, pelagic thresher, silky, and spinner shark. However, for the late-maturing species such as dusky, sandbar, shortfin mako, oceanic whitetip, bigeye thresher, and pelagic thresher shark, a reduce of catch or TAC management measure has been suggested [57]. Tsai et al. [30,31] demonstrated that the bigeye thresher stock in the Northwest Pacific was declining in population size under current fishery condition and overfishing was likely occurring. Similar conclusion was also made for the shortfin mako shark [32,33,34,43] using various approaches. The pelagic thresher and female smooth hammerhead shark also have been reported as overfishing at current fishing effort [28,29,35]. Based on aforementioned single species assessment results, it is suggested, for precautionary purpose, the management plan for each species should be developed.

In addition to the conventional PSA, quantitative ERA tools such as sustainability assessment for fishing effects (SAFE) [89,90,91,92,93] and Ecological Assessment of Sustainable Impacts of Fisheries (EASI-Fish) [94,95] have been used to derive a proxy for fishing mortality (F) based on the productivity and susceptibility of fish. The F values estimated from SAFE were comparable to those derived from data-rich quantitative stock assessments in most cases although overestimation of F may occur [78]. Therefore, the SAFE should be considered in future ecological risk assessment to provide more solid recommendations for management measures.

## 5. Conclusions

The integrated ERA method developed in this study can prioritize the risk of pelagic sharks in the western North Pacific. However, this approach cannot provide concrete management information such as total allowable catch (TAC), biological reference points (BRPs), and optimum fishing effort until further quantitative ERA is conducted. Even though the integrated ERA cannot replace the conventional stock assessment method, it can provide useful information for precautionary management measures. In addition to the ban retention on the silky and oceanic whitetip shark, for the species in high risk group (groups 1 and 2), stock assessment as well as rigorous management measures such as catch quota, and size limit are recommended. Setting total allowable catch quota is recommended for the species in group 3 and a consistent monitoring scheme is suggested for the species in groups 4 and 5. The integrated ERA should be updated regularly according to the availability of new information of the productivity and susceptibility of sharks.

## Figures and Tables

**Figure 1 animals-11-02161-f001:**
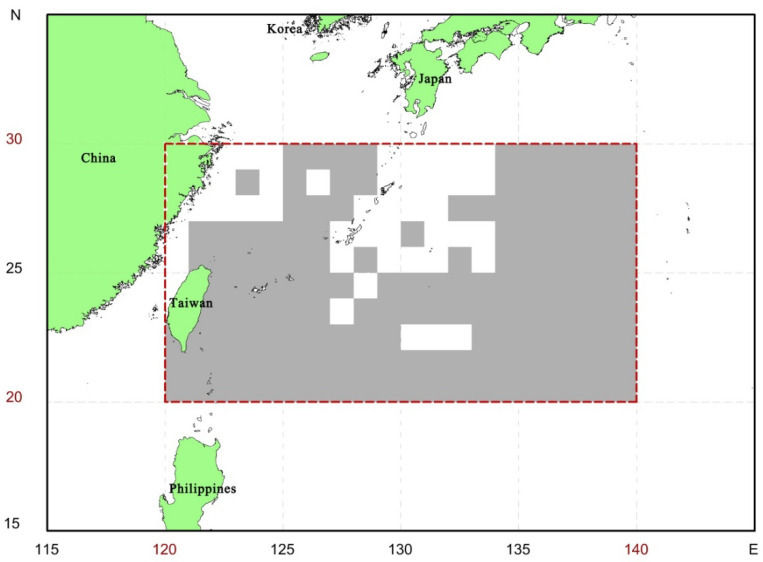
The study area (from 20° N to 30° N, 120° E to 140° E) in the present study. Gray areas indicate fishing grounds of the Taiwanese coastal and offshore longline fishing vessels.

**Figure 2 animals-11-02161-f002:**
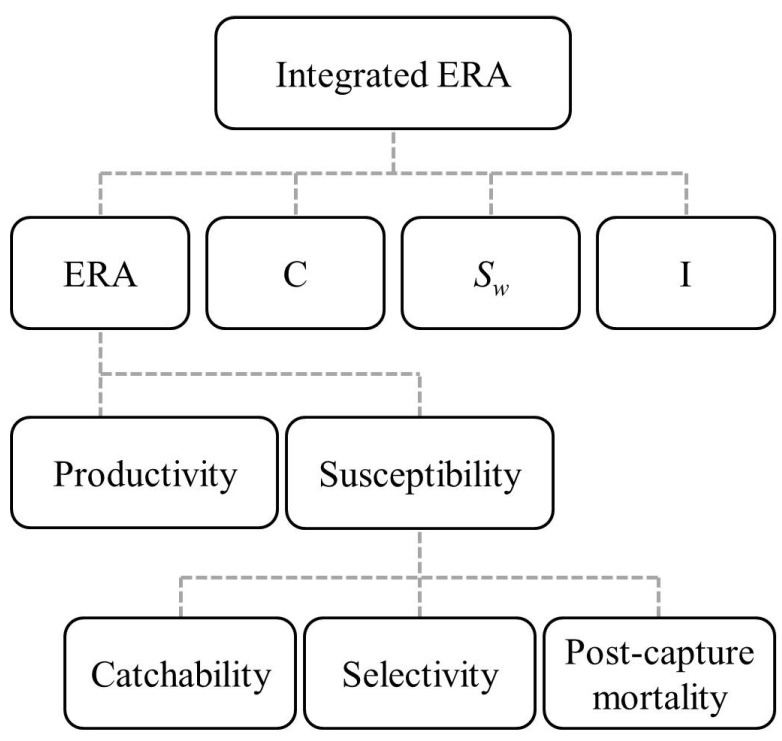
The flow chart of integrated ERA methods used in this study. Four indices were used to conduct the integrated assessment on 10 shark species: ecological risk assessment (ERA), IUCN Red List (C), body weight variation trend ($S_{W}$), and reflection point of population growth curve (I).

**Figure 3 animals-11-02161-f003:**
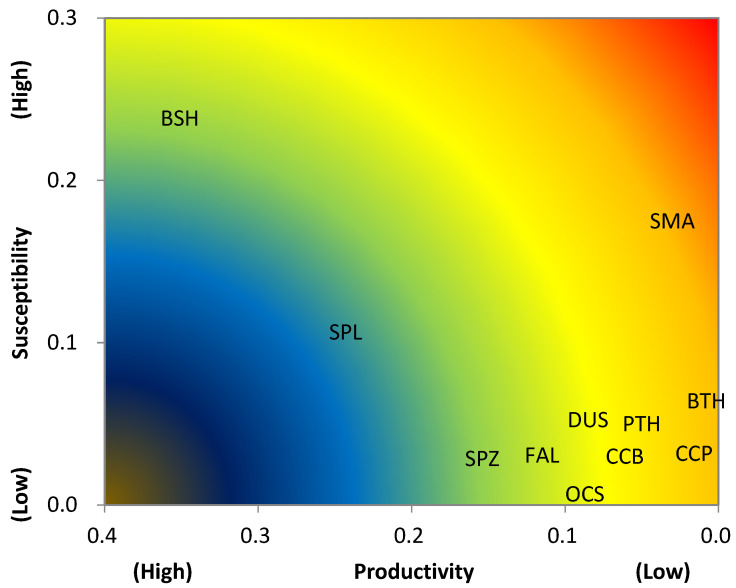
Productivity and susceptibility plot for 11 pelagic shark species in the western North Pacific Ocean. Productivity is expressed as r (intrinsic rate of population growth) and susceptibility as the catchability, selectivity, and post-capture mortality. PTH: pelagic thresher, BTH: bigeye thresher, CCB: spinner, FAL: silky, OCS: oceanic whitetip, DUS: dusky, CCP: sandbar, SMA: shortfin mako, BSH: blue, SPL: scalloped hammerhead, SPZ: smooth hammerhead.

**Figure 4 animals-11-02161-f004:**
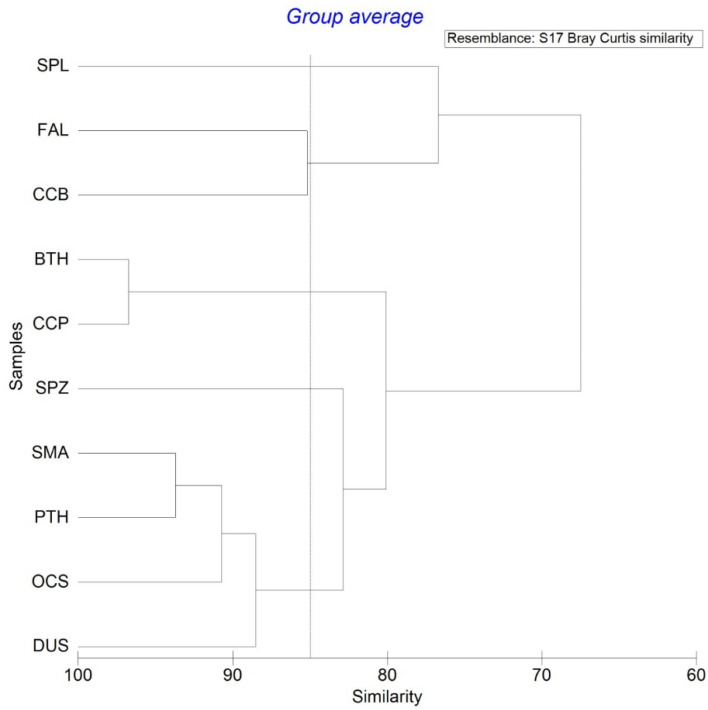
Results of the cluster analysis of the integrated ecological risk assessment for the 10 pelagic shark species in the western North Pacific Ocean. The scalloped hammerhead (SPL) has the highest risk (group 1), followed by silky (FAL) and spinner (CCB) (group 2). The bigeye thresher (BTH) and sandbar shark (CCP) fall in the group 3. Group 4 includes smooth hammerhead (SPZ), and group 5 includes shortfin mako (SMA), pelagic thresher (PTH), oceanic whitetip (OCS), and dusky shark (DUS).

**Figure 5 animals-11-02161-f005:**
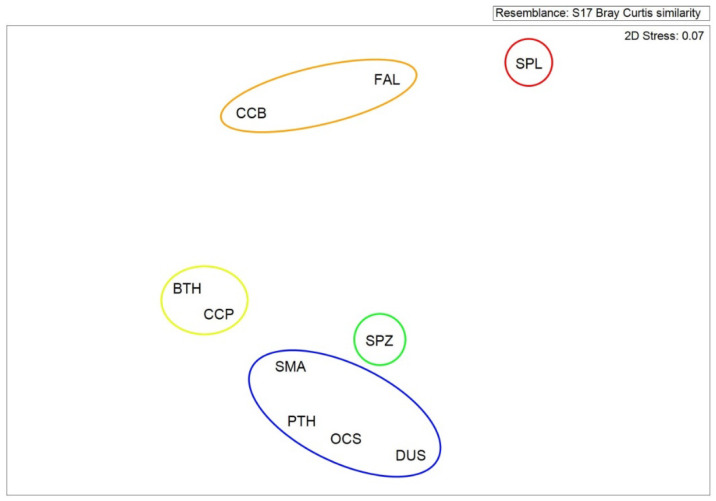
Results of the non-parametric multi-dimensional scaling (NMDS) analysis of the integrated ecological risk assessment for the 10 pelagic shark species in the western North Pacific Ocean. PTH: pelagic thresher, BTH: bigeye thresher, CCB: spinner, FAL: silky, OCS: oceanic whitetip, DUS: dusky, CCP: sandbar, SMA: shortfin mako, SPL: scalloped hammerhead, SPZ: smooth hammerhead.

**Table 1 animals-11-02161-t001:** Age and growth parameters of 11 shark species in the western North Pacific Ocean.

Species	L_max_ (cm TL)	L_∞_ (cm)	k (Year^−1^)	t_0_	t_max_ (Year)	References
Pelagic thresher	365.18	382.94	0.09	−7.67	27.57	[36]
Bigeye thresher	422.00	422.00	0.09	−4.21	28.35	[37]
Spinner	274.00	288.20	0.15	−1.99	17.85	[38]
Silky	256.00	332.00	0.08	−2.76	32.99	[39]
Oceanic whitetip	268.00	323.80	0.11	−0.37	12.30	[40]
Dusky	364.00	415.70	0.06	−3.42	50.08	[41]
Sandbar	210.00	210.00	0.17	−2.30	15.32	[42]
Shortfin mako	375.00	413.80	0.05	—	40.04	[43,44]
Scalloped hammerhead	324.00	319.72	0.25	−0.41	11.62	[45]
Smooth hammerhead	324.00	375.20	0.11	−1.31	25.73	[46]
Blue	323.00	322.70	0.16	−1.33	17.24	[47]

L_max_: maximum observed length, L_∞_: asymptotic length, k: growth coefficient, t_0_: age at length 0, t_max_: longevity. Pelagic thresher, *Alopias pelagicus*; Bigeye thresher, *A. superciliosus*; Spinner, *Carcharhinus brevipinna*; Silky, *C. falciformis*; Oceanic whitetip, *C. longimanus*; Dusky shark, *C. obscurus*; Sandbar, *C. plumbeus*; Shortfin mako, *Isurus oxyrinchus*; Blue, *Prionace glauca*; Scalloped hammerhead, *Sphyrna lewini*; Smooth hammerhead shark, *S. zygaena*.

**Table 2 animals-11-02161-t002:** Reproductive parameters of 11 pelagic shark species in the western North Pacific Ocean.

Species	R	L_b_ (cm)	L_m_ (cm)	t_m_ (Year)	f	G_p_ (Month)	R_c_ (Year)	References
Pelagic thresher	av	174.0	287.0	8.6	2	12	1	[36]
Bigeye thresher	av	148.7	336.6	12.85	2	12	1	[49]
Spinner	v	67.5	222.5	7.8	8.5	11	2	[38]
Silky	v	69.5	215.0	9.7	9	12	2	[39]
Oceanic whitetip	v	64.0	194.7	8.23	10	12	2	[40]
Dusky	v	101.0	281.0	16.4	11	13	2	[41]
Sandbar	v	62.5	172.5	7.85	7.5	11	2	[50]
Shortfin mako	av	74.0	278.0	20	11.1	24	3	[43,44]
Scalloped hammerhead	v	48.5	230.0	4.7	25.8	10	2	[51]
Smooth hammerhead	v	55.0	259.4	11	30	10	2	[52]
Blue	v	45.0	189.0	4.2	29	10	2	[53]

R: reproductive strategy (av: aplacental viviparity, v: viviparity), L_b_: size at birth, L_m_: size at maturity, t_m_: age at maturity, f: littler size, G_p_: gestation period, R_c_: reproductive cycle, sex ratio of embryos (F/(F + M)) was set as 0.5.

**Table 3 animals-11-02161-t003:** IUCN red list global status of 11 shark species in the western North Pacific Ocean. The IUCN Red List index (C) indicates the relative risk of species. LC = 0.2, NT = 0.4, VU = 0.6, EN = 0.8 and CR = 1.0. (IUCN [55]).

Species	IUCN Status	IUCN Status	IUCN Red List Index (C)
Pelagic thresher	Endangered	EN	0.8
Bigeye thresher	Vulnerable	VU	0.6
Spinner	Near Threatened	NT	0.4
Silky	Vulnerable	VU	0.6
Oceanic whitetip	Critically Endangered	CR	1.0
Dusky	Endangered	EN	0.8
Sandbar	Vulnerable	VU	0.6
Shortfin mako	Endangered	EN	0.8
Scalloped hammerhead	Critically Endangered	CR	1.0
Smooth hammerhead	Vulnerable	VU	0.6
Blue	Near Threatened	NT	0.4

**Table 4 animals-11-02161-t004:** Estimated demographic parameters for 11 pelagic shark species in the western North Pacific Ocean without fishing mortality.

Species	r	λ	t_x2_	G
Pelagic thresher	0.050	1.051	13.9	14.3
Bigeye thresher	0.008	1.008	82.6	18.0
Spinner	0.061	1.063	11.3	11.1
Silky	0.115	1.122	6.0	16.4
Oceanic whitetip	0.087	1.091	7.9	10.7
Dusky	0.085	1.088	8.2	26.5
Sandbar	0.016	1.016	43.5	10.5
Shortfin mako	0.030	1.031	23.0	26.7
Scalloped hammerhead	0.243	1.275	2.9	7.1
Smooth hammerhead	0.154	1.166	4.5	15.6
Blue	0.351	1.420	2.0	8.5

r: finite rate of population increase; λ: population increase rate; $t_{x2}$: population doubling time; G: generation time in years.

**Table 5 animals-11-02161-t005:** Parameters of susceptibility in the ecological risk assessment of the 11 shark species in the Northwest Pacific pelagic species. The susceptibility indicators (S) reflect the risk of fisheries development.

Species	Catchability	Selectivity	Post-Capture Mortality	Susceptibility
Pelagic thresher	0.0647	1.0000	0.7800	0.0505
Bigeye thresher	0.0827	1.0000	0.7800	0.0645
Spinner	0.0396	0.9970	0.7700	0.0304
Silky	0.0432	0.9960	0.7230	0.0311
Oceanic whitetip	0.0108	0.9350	0.7200	0.0073
Dusky	0.0576	1.0000	0.9200	0.0529
Sandbar	0.0396	0.9480	0.8600	0.0323
Shortfin mako	0.1906	1.0000	0.9200	0.1754
Scalloped hammerhead	0.1331	0.9680	0.8300	0.1069
Smooth hammerhead	0.0360	0.9490	0.8500	0.0290
Blue	0.3022	1.0000	0.7900	0.2387

**Table 6 animals-11-02161-t006:** Ecological risk assessments of 11 pelagic sharks in the western North Pacific Ocean using the Euclidean distance (D).

Species	Productivity	Susceptibility	ERA (D)	Risk Rank
Pelagic thresher	0.0500	0.0505	0.4528	4
Bigeye thresher	0.0080	0.0645	0.4962	2
Spinner	0.0610	0.0304	0.4400	5
Silky	0.1150	0.0311	0.3863	8
Oceanic whitetip	0.0870	0.0073	0.4131	7
Dusky	0.0850	0.0529	0.4184	6
Sandbar	0.0160	0.0323	0.4851	3
Shortfin mako	0.0300	0.1754	0.5017	1
Scalloped hammerhead	0.2430	0.1069	0.2784	11
Smooth hammerhead	0.1540	0.0290	0.3472	9
Blue	0.3510	0.2387	0.2814	10

**Table 7 animals-11-02161-t007:** Integrated ecological risk assessment of 11 pelagic sharks in the western North Pacific Ocean. ERA: ecological risk assessment; C: IUCN Red List index; *Sw*: index of body weight variation trend, inside parenthesis is the standard error (SE); I: position of the inflection point of the population growth curve. The integrated ERA did not include the blue shark as its *Sw* was not available.

Species	ERA	C	*S_W_*	I
Pelagic thresher	0.4528	0.8	−0.0828 (0.1084)	0.6955
Bigeye thresher	0.4962	0.6	−0.4360 (0.0788)	0.9870
Spinner	0.4400	0.4	−0.9410 (0.4563)	0.7050
Silky	0.3863	0.6	−1.2762 (0.2110)	0.5139
Oceanic whitetip	0.4131	1.0	−0.1319 (0.0781)	0.6454
Dusky	0.4184	0.8	−0.0163 (0.3635)	0.4820
Sandbar	0.4851	0.6	−0.3079 (0.1383)	0.9669
Shortfin mako	0.5017	0.8	−0.2818 (0.1477)	0.6740
Scalloped hammerhead	0.2784	1.0	−1.8279 (0.1894)	0.5323
Smooth hammerhead	0.3472	0.6	−0.2404 (0.0860)	0.4697
Blue	0.2814	0.4	---	0.4286

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
