# Peer review of "Vulnerability Assessment of Pelagic Sharks in the Western North Pacific by Using an Integrated Ecological Risk Assessment"

_animals, 2021, doi:10.3390/ani11082161_

Round 1

Reviewer 1 Report

This manuscript developed an integrated ecological risk assessment (ERA) to evaluate the vulnerability of 11 shark species in the western North Pacific using the productivity of species and their susceptibility to fisheries.  The paper is interesting and potentially useful, as it demonstrated the potential impact of longline fishery on pelagic shark species in the western North Pacific, and made an attempt to use the results to provide suggestions for the management plan and environmental policies of the study area.  I think that the combined use of the intrinsic rate of population increase, catchability, selectivity, post-capture mortality and three indices (i.e., IUCN Red List category, body weight variation trend, and the inflection point of population growth curve coupled with ERA) in a multivariate statistical modeling framework is an interesting advantage of this paper in relation to others that deal with similar topics.  There are some justifications the authors need to include, which will enrich the content of the research while clarifying the selection and implementation of the approaches used.  The specific comments are:

Summary

  • (1) Page 1 Line 13.  “integrated Ecological risk assessment (ERA)” to “integrated ecological risk assessment (ERA)”.

Introduction

  • (2) Some information such as background on pelagic shark populations in the study area would be better in the Introduction section, to set the stage.  I would also suggest emphasizing somewhere in the Introduction section the significance of the combined use of the intrinsic rate of population increase, catchability, selectivity, post-capture mortality and three indices (i.e., IUCN Red List category, body weight variation trend, and the inflection point of population growth curve coupled with ERA) in a multivariate statistical modeling framework, because these approaches might help people from broader field find your work useful rather than just people working on the particular area or species being interested in it.

Materials and Methods

  • (3) Page 3 Lines 112-114.  “Species-specific landing data including the whole weight of each individual fish (except very few small-size individuals were in bulk weight) at the Nanfangao fishing port, northeastern Taiwan from 1989-2011 were collected from the sales records” to “Species-specific landing data, including the whole weight of each individual fish (except very few small-size individuals were in bulk weight) at the Nanfangao fishing port, northeastern Taiwan from 1989-2011, were collected from the sales records”.
  • (4) Page 6 Lines 181-182.  There is little explanation as to why cluster analysis or non-parametric multi-dimensional scaling (NMDS) was chosen.  There have been some other modeling approaches that were used in fishery ecology when it comes to predicting species groups.  However, there is little justification as to why the authors only chose cluster analysis or NMDS.  I would suggest adding a brief explanation to justify why these methods and not other also commonly used approaches were chosen.  Given the potential of modeling approaches to influence the management plan and environmental policies of the study area, it would be better to critically assess why some models are chosen and their potential and weaknesses to the specific environment considered.
  • (5) Page 6 Line 182.  When you used cluster analysis coupled with NMDS to group the ten species based on their similarity of integrated ERA, did you use raw or transformed values from the integrated ERA to apply the modeling approaches on?  It would be better to specify a little more even though data transformation may not be necessary for some situations, because it is an important step to make sure the variables meet the underlying assumptions of the algorithms before conducting any statistical analyses.  Statistical approaches with different mechanisms may have different underlying assumptions of normality, linearity or multicollinearity, and some variables may need to be transformed to meet specific assumptions.

Discussion

  • (6) Page 12 Lines 324-326.  I would suggest specifying which part of your results supports that “the results of integrated ERA did not change even these uncertainties have been taken into account”.

Tables and Figures

  • (7) Table 1 Lines 122-123.  I would suggest making the upper or lower case of the abbreviations consistent throughout your manuscript (e.g., K or k, Tmax or tmax).
  • (8) Table 3 Line 174.  “IUCN Red List index (C) indicate” to “IUCN Red List index (C) indicates”.
  • (9) Table 7.  I would suggest switching the columns of Sw and C in this table to make it consistent throughout the manuscript.
  • (10) Figure 5 Line 270.  “the Multi-Dimensional Scaling (MDS) analysis” to “the non-parametric multi-dimensional scaling (NMDS) analysis”.

Author Response

This manuscript developed an integrated ecological risk assessment (ERA) to evaluate the vulnerability of 11 shark species in the western North Pacific using the productivity of species and their susceptibility to fisheries.  The paper is interesting and potentially useful, as it demonstrated the potential impact of longline fishery on pelagic shark species in the western North Pacific, and made an attempt to use the results to provide suggestions for the management plan and environmental policies of the study area.  I think that the combined use of the intrinsic rate of population increase, catchability, selectivity, post-capture mortality and three indices (i.e., IUCN Red List category, body weight variation trend, and the inflection point of population growth curve coupled with ERA) in a multivariate statistical modeling framework is an interesting advantage of this paper in relation to others that deal with similar topics.  There are some justifications the authors need to include, which will enrich the content of the research while clarifying the selection and implementation of the approaches used.  The specific comments are:

Summary

  • (1) Page 1 Line 13.  “integrated Ecological risk assessment (ERA)” to “integrated ecological risk assessment (ERA)”.

 Response: The editorial correction has been made.

Introduction

  • (2) Some information such as background on pelagic shark populations in the study area would be better in the Introduction section, to set the stage.  I would also suggest emphasizing somewhere in the Introduction section the significance of the combined use of the intrinsic rate of population increase, catchability, selectivity, post-capture mortality and three indices (i.e., IUCN Red List category, body weight variation trend, and the inflection point of population growth curve coupled with ERA) in a multivariate statistical modeling framework, because these approaches might help people from broader field find your work useful rather than just people working on the particular area or species being interested in it.

Response: We have added some sentences to describe the background on pelagic sharks in the study area in Introduction section. In addition, the significance of using an integrated ERA to assess the vulnerability of pelagic sharks has also been added in Introduction section.

Materials and Methods

  • (3) Page 3 Lines 112-114.  “Species-specific landing data including the whole weight of each individual fish (except very few small-size individuals were in bulk weight) at the Nanfangao fishing port, northeastern Taiwan from 1989-2011 were collected from the sales records” to “Species-specific landing data, including the whole weight of each individual fish (except very few small-size individuals were in bulk weight) at the Nanfangao fishing port, northeastern Taiwan from 1989-2011, were collected from the sales records”.
  • Response: This sentence has been revised based on the comment.

  • (4) Page 6 Lines 181-182.  There is little explanation as to why cluster analysis or non-parametric multi-dimensional scaling (NMDS) was chosen.  There have been some other modeling approaches that were used in fishery ecology when it comes to predicting species groups.  However, there is little justification as to why the authors only chose cluster analysis or NMDS.  I would suggest adding a brief explanation to justify why these methods and not other also commonly used approaches were chosen.  Given the potential of modeling approaches to influence the management plan and environmental policies of the study area, it would be better to critically assess why some models are chosen and their potential and weaknesses to the specific environment considered.

Response: We have added some text to describe the reason why we chose cluster analysis and NMDS because these methods have been used in elasmobranch management based on life history analysis.

  • (5) Page 6 Line 182.  When you used cluster analysis coupled with NMDS to group the ten species based on their similarity of integrated ERA, did you use raw or transformed values from the integrated ERA to apply the modeling approaches on?  It would be better to specify a little more even though data transformation may not be necessary for some situations, because it is an important step to make sure the variables meet the underlying assumptions of the algorithms before conducting any statistical analyses.  Statistical approaches with different mechanisms may have different underlying assumptions of normality, linearity or multicollinearity, and some variables may need to be transformed to meet specific assumptions.

Response: As these indices were independent, not normally distributed with small sample size (n = 10), a non-parametric multi-dimensional scaling (NMDS) was used in grouping. The raw values of the four indices except Sw which was taking absolute value (positive) were used in the NMDS analysis. 

Discussion

  • (6) Page 12 Lines 324-326.  I would suggest specifying which part of your results supports that “the results of integrated ERA did not change even these uncertainties have been taken into account”.

 Response: Even aforementioned uncertainties in life history parameters were taken account in our analyses as different scenarios, the ecological risk derived from the integrated ERA did not change suggesting that our results were robust.

Tables and Figures

  • (7) Table 1 Lines 122-123.  I would suggest making the upper or lower case of the abbreviations consistent throughout your manuscript (e.g., K or k, Tmax or tmax).
  • (8) Table 3 Line 174.  “IUCN Red List index (C) indicate” to “IUCN Red List index (C) indicates”.
  • (9) Table 7.  I would suggest switching the columns of Sw and C in this table to make it consistent throughout the manuscript.
  • (10) Figure 5 Line 270.  “the Multi-Dimensional Scaling (MDS) analysis” to “the non-parametric multi-dimensional scaling (NMDS) analysis”.

Response: These corrections have been made.

Reviewer 2 Report

I think this is an interesting paper. In table 1 and 2 some scientific names are written in full and others in abbreviated form. I strongly suggest that authors standardize the style and write them all in abbreviated form.

Author Response

We have revised all Tables and use common names for all species to match the usage in the text. To assist readers to understand the species, a list of common names, scientific names, and species code was added in the description of Table 1. 

Reviewer 3 Report

General comments:  

This is a useful paper, based on extensive previous work, and will be valuable to managers and of general interest to those involved in shark conservation.

There are numerous small errors of English in this text, mostly of grammar. All of these slow a reader in terms of rapidly following the meaning, but some errors make the meaning unclear. I have in general focused my comments only on these latter errors. It would make the paper much easier for the audience to read and very probably increase the readership, if the text was carefully edited for all these minor errors.

The ERA is based on extensive work on growth and life history data of the species concerned in the area of the fishery, which is very impressive. The equation for R0 however, assumes not only that the sex ratio of embryos is 1:1, but also that the estimates mortality and growth rates for each species represent females - that is, if both sexes were used in the estimation, the assumption is that male and female growth and mortality rates are equal. This assumption should be checked in the references used, and then discussed in the discussion section on assumptions.

The issue of how much of the range of the species is fished by this fishery is briefly covered in the discussion, but should be more carefully considered. The risk that this fishery poses to each species will depend not only on how easily the species is caught by the longline method used in this fishery, but also on how much of the species’ range is fished. The overall risks to these species will also depend on the extent to which other fisheries also inflict mortality on these species. This might explain some strong disparities between the ERA results and the IUCN estimated risks. Again, these issues deserve discussion.

The use of the species codes in the 7 tables does not seem useful. It requires first checking back to the first two tables to determine which sharks are denoted by the codes, and then relating the common names used in the text to the scientific names.  There is space in the tables to use the scientific names, and these would make the information easier to follow for the average reader, especially if the text also used the scientific names. If the text is to use the common names throughout, then these should be added to Table 1 in place of the species codes. At present there is only an incomplete list in the text (lines 103-106) that leaves three species for the reader to guess the scientific name and code. The codes are clearly useful for Figure 3 and the supplementary figures, but a good solution would be to include both scientific names and codes in table 6, and place this table close to the figure (as in the MS); and to put a key to both common names and scientific names in the supplementary material.

Specific comments:

I don’t understand the distinction between Summary and Abstract, and it seems these authors do not either, as they look remarkably similar. At the end of the Abstract, the text should read: “Rigorous management measures…” not “A rigorous …”

The application of ERAs for fishery vulnerability needs to be understood in the context of the particular fishery considered. Here, the summary mentions a longline fishery, the Abstract does not mention any fishery, and the Introduction initially implies that the focus is on tuna fisheries, before we eventually learn the study evaluates the vulnerabilities due to the Taiwanese offshore longline fishery in the western North Pacific. This should be stated in the Abstract.

Lines 70-71: ERA methods estimate the vulnerability of target and bycatch species stocks TO fishing gear, and it is not a “stock’s vulnerability”

Lines 86-89: the two sentences here should be one sentence, with a comma between them, to make sense.

Lines 112-3: The meaning of “except very few small-size individuals were in bulk weight” is not clear. Does this mean that some small sized individuals were weighed in groups, so that only average weights are available? The meaning should be more clearly explained.

Lines 112-114: The text of this sentence should be re-organised. “Species-specific landing data from 1989-2011 were collected from the sales records at the Nanfangao fishing port, northeastern Taiwan. These records included the whole weight of each individual fish (except that …”

Lines 138-9: More correctly, Productivity is the ability of the population to withstand the fishery exploitation by replacing individuals through reproduction, survival and growth of individuals.

Lines 140-1: Similarly, Susceptibility is the impact of a fishery on a species population.

Line 149: “post-mortality” should be “post-capture mortality”.

Line 167: “threshed” should be “threatened”.

Lines 179-80: As the blue shark is processed at sea, the reader is left confused at this point in the paper as to how its catch weight, individual weights and other data were determined from the landing data to calculate parameters for the ERA. The explanation in lines 292-295 of the discussion should be presented here. Further, the inflection point of the population growth curve and the IUCN red list category for the blue shark could be presented in the text in sections 3.4 and 3.6, so that readers can assess where it is likely to fit into the ultimate assessment using the integrated ERA.

Table 4 key: “population increase rate” should be: “finite rate of population increase”;  “time of population become twice” should be: “population doubling time”; “generation length in year” should be: “generation time in years”.

Figure 3 caption: The caption should refer to where the species codes are explained. Also, susceptibility should be defined as the product of catchability, selectivity and post-capture mortality.

Table 7: The slopes of the declines in average weight over the period should be presented in the table with the standard error of the slope, so that readers can judge the confidence limits around these declines. If possible, the inflexion point of the population growth curves should also have a standard error shown.

Lines 246-7: The supplementary figures show clearly that the oceanic whitetip did not have a small decline in size, it was the Dusky shark where the decline was small. In addition, the colours used to show the annual data in the supplementary figures mean that the line for C. plumbeus (CCP) is hard to see. All these lines do not need varied colours.

Line 250: “near threshed” should be “near threshold category”.

Figure 4: The rationale for the vertical line denoting which species were grouped should be explained in the figure caption.

Line 340: Are ALL these shark species highly migratory, so that individuals would move in an out of this very large fishing zone? If some are less likely to move or have smaller ranges, this should be considered.

Line 345: I would assume that sharks are attracted from some distance to baits on longlines, whether these distances are vertical or horizontal.

Lines 348-352: Catchability is usually measured by considering how much of a species is caught versus the fishing effort. In this case, the percentage of each species in the total catch weight in landings from 1989-2011 (lines 141-3) is a measure of relative catchability. It would change if the abundance of other species in the catch changes, and as there are only 11 species considered, the catchability of each species is sensitive to how much of any other species is caught. Some of these species have declined substantially over the period, changing the estimated catchability of others. This issue with the method used should be discussed here. 

Lines 395-425: The ERA and Integrated ERA results here are in stark contrast to the IUCN ratings for several species: Carcharhinus longimanus (oceanic whitetip, OCS) and Sphyrna lewini (scalloped hammerhead, SPL) are critically endangered on the IUCN list, but both low on the ERA list, and the first is also in the group of least concern for the IERA list. Alopias pelagicus (pelagic thresher, PTH), C. obscurus (dusky shark, DUS) and Isurus oxyrinchus (shortfin mako shark, SMA) are in the lowest risk group in the ERA, yet all are recorded as endangered (C=0.8) on the IUCN list.

    The Authors discuss Sphyrna lewini but do not explain why their ERA would produce such a low risk even though the sizes of this species caught have declined dramatically over the period of the fishery considered. As suggested by my comment (lines 348-352) on the relative catchability parameter, is this perhaps because the depletion of the species has led to much lower catches in recent times, so that the relative catchability of this species has been under-estimated by their method? Selectivity, Post-capture mortality and Susceptibility are all relatively high (Table 5).

    The ERA for Carcharhinus falciformis is also a low value, although selectivity is high. The authors state the IERA ranking is high due to the decline in median weight over the fishing period. Could the low ERA also be due to an underestimate of the catchability due to depletion of this species?

    The authors also discuss their ERA rankings versus other assessments of Alopias superciliosus and Isurus oxyrinchus. But I suggest the disparity in assessment for Carcharhinus longimanus, Alopias pelagicus, and C.obscurus, all should be explained. It is important to investigate whether the assessment used here might underestimate risks for these species.

Lines 427-432:  Management has to balance the collection of more data with costs. It would therefore be more useful to point out which kinds of data are most useful to gain a good assessment (or rsk analysis), than to say simply that biological and fisheries data should be updated.

The conclusion makes some very good points.

I have skimmed through some of the references and noted some errors, which suggest that a thorough check of the references is needed.

References 4 & 13: The journal Science is not usually abbreviated to “Sci”.

Refs 5 & 9: Similarly, Nature is not usually abbreviated.

Ref 7. “Fish” is an abbreviation of what? Fisheries? Or Fish and Fisheries?

Ref 15: ‘de Bruyn’ is the family name of the second author.

Ref 25 has the first names of authors rather than their last names.

Author Response

General comments:  

This is a useful paper, based on extensive previous work, and will be valuable to managers and of general interest to those involved in shark conservation.

There are numerous small errors of English in this text, mostly of grammar. All of these slow a reader in terms of rapidly following the meaning, but some errors make the meaning unclear. I have in general focused my comments only on these latter errors. It would make the paper much easier for the audience to read and very probably increase the readership, if the text was carefully edited for all these minor errors.

Response: In addition to the revision based on the Reviewer’s comments on grammatical errors, we have checked the text carefully and made some additional corrections and hopefully this revised MS will be more readable.

The ERA is based on extensive work on growth and life history data of the species concerned in the area of the fishery, which is very impressive. The equation for R0 however, assumes not only that the sex ratio of embryos is 1:1, but also that the estimates mortality and growth rates for each species represent females - that is, if both sexes were used in the estimation, the assumption is that male and female growth and mortality rates are equal. This assumption should be checked in the references used, and then discussed in the discussion section on assumptions.

Response: The estimation of intrinsic rate of population growth (r) in demographic analysis was based on the assumption that males are not the determining factor affecting the population growth. Thus, the estimation was based on the life history parameters of females. We have added the following text in Discussion section. “The intrinsic rate of population growth (r), estimated from demographic analysis, was used as productivity in this study. However, uncertainties of life history parameters and natural mortality that may lead to biased estimation of r [61] using this approach. Therefore, stochastic approach should be considered to obtain a more robust estimate of intrinsic population growth rate in the future.”

The issue of how much of the range of the species is fished by this fishery is briefly covered in the discussion, but should be more carefully considered. The risk that this fishery poses to each species will depend not only on how easily the species is caught by the longline method used in this fishery, but also on how much of the species’ range is fished. The overall risks to these species will also depend on the extent to which other fisheries also inflict mortality on these species. This might explain some strong disparities between the ERA results and the IUCN estimated risks. Again, these issues deserve discussion.

Response: We agreed that other fishery may have impact on the 11 shark species population. However, the majority of these shark catch in the study area was from longline fishery with very minor from drift net fishery which can be ignored. As for the range of distribution for some species may be outside the study area such as the blue shark, shortfin mako, and oceanic whitetip, we have added some text in Discussion section to touch this point. We also added some sentences to discuss possible reasons causing the difference between our estimate and that of IUCN in Discussion section. The different study scale (regional vs global) and different approaches (methodologies) may be the two key factors.

The use of the species codes in the 7 tables does not seem useful. It requires first checking back to the first two tables to determine which sharks are denoted by the codes, and then relating the common names used in the text to the scientific names.  There is space in the tables to use the scientific names, and these would make the information easier to follow for the average reader, especially if the text also used the scientific names. If the text is to use the common names throughout, then these should be added to Table 1 in place of the species codes. At present there is only an incomplete list in the text (lines 103-106) that leaves three species for the reader to guess the scientific name and code. The codes are clearly useful for Figure 3 and the supplementary figures, but a good solution would be to include both scientific names and codes in table 6, and place this table close to the figure (as in the MS); and to put a key to both common names and scientific names in the supplementary material.

Response: Thanks for the suggestions. The species code used in this study is commonly used in regional fisheries management organizations (RFMOs) such as ICCAT, WCPFC, and IOTC as well as the shark literature. To increase the readability for general readers, we have added a list of species (common) names, scientific names, and species code in the legends of Table 1 and Supplement Fig. 1 for reference. As common names were used throughout the MS, only common names were used in Tables 1-7.

Specific comments:

I don’t understand the distinction between Summary and Abstract, and it seems these authors do not either, as they look remarkably similar. At the end of the Abstract, the text should read: “Rigorous management measures…” not “A rigorous …”

The application of ERAs for fishery vulnerability needs to be understood in the context of the particular fishery considered. Here, the summary mentions a longline fishery, the Abstract does not mention any fishery, and the Introduction initially implies that the focus is on tuna fisheries, before we eventually learn the study evaluates the vulnerabilities due to the Taiwanese offshore longline fishery in the western North Pacific. This should be stated in the Abstract.

Response: Summary is required for “Animals” and I tried to differentiate it with Abstract in this revised version. Description of tuna longline fishery has been added in Abstract and editorial error has been corrected.

Lines 70-71: ERA methods estimate the vulnerability of target and bycatch species stocks TO fishing gear, and it is not a “stock’s vulnerability”

Response: This wording has been revised.

Lines 86-89: the two sentences here should be one sentence, with a comma between them, to make sense.

Response: This sentence has been revised accordingly.

Lines 112-3: The meaning of “except very few small-size individuals were in bulk weight” is not clear. Does this mean that some small sized individuals were weighed in groups, so that only average weights are available? The meaning should be more clearly explained.

Lines 112-114: The text of this sentence should be re-organised. “Species-specific landing data from 1989-2011 were collected from the sales records at the Nanfangao fishing port, northeastern Taiwan. These records included the whole weight of each individual fish (except that …”

Response: This sentence has been changed as “Species-specific landing data, including the whole weight of each individual fish (except very few small-size individuals that were weighed in group and only average weights were available) at the Nanfangao fishing port, northeastern Taiwan from 1989-2011, were collected from the sales records.”

Lines 138-9: More correctly, Productivity is the ability of the population to withstand the fishery exploitation by replacing individuals through reproduction, survival and growth of individuals.

Response: This sentence has been revised accordingly.

Lines 140-1: Similarly, Susceptibility is the impact of a fishery on a species population.

Response: This wording has been revised as suggested.

Line 149: “post-mortality” should be “post-capture mortality”.

Line 167: “threshed” should be “threatened”.

Response: The typos have been corrected.

Lines 179-80: As the blue shark is processed at sea, the reader is left confused at this point in the paper as to how its catch weight, individual weights and other data were determined from the landing data to calculate parameters for the ERA. The explanation in lines 292-295 of the discussion should be presented here. Further, the inflection point of the population growth curve and the IUCN red list category for the blue shark could be presented in the text in sections 3.4 and 3.6, so that readers can assess where it is likely to fit into the ultimate assessment using the integrated ERA.

Response: We have moved text in Discussion section to Materials and Methods section. The ERA, the inflection point of the population growth curve and the IUCN red list category for the blue shark have been presented in Table 7. So, readers can make their judgement based on this information.

Table 4 key: “population increase rate” should be: “finite rate of population increase”;  “time of population become twice” should be: “population doubling time”; “generation length in year” should be: “generation time in years”.

Response: These editorial changes have been made.

Figure 3 caption: The caption should refer to where the species codes are explained. Also, susceptibility should be defined as the product of catchability, selectivity and post-capture mortality.

Table 7: The slopes of the declines in average weight over the period should be presented in the table with the standard error of the slope, so that readers can judge the confidence limits around these declines. If possible, the inflexion point of the population growth curves should also have a standard error shown.

Lines 246-7: The supplementary figures show clearly that the oceanic whitetip did not have a small decline in size, it was the Dusky shark where the decline was small. In addition, the colours used to show the annual data in the supplementary figures mean that the line for C. plumbeus (CCP) is hard to see. All these lines do not need varied colours.

Line 250: “near threshed” should be “near threshold category”.

Figure 4: The rationale for the vertical line denoting which species were grouped should be explained in the figure caption.

Line 340: Are ALL these shark species highly migratory, so that individuals would move in an out of this very large fishing zone? If some are less likely to move or have smaller ranges, this should be considered.

Line 345: I would assume that sharks are attracted from some distance to baits on longlines, whether these distances are vertical or horizontal.

Lines 348-352: Catchability is usually measured by considering how much of a species is caught versus the fishing effort. In this case, the percentage of each species in the total catch weight in landings from 1989-2011 (lines 141-3) is a measure of relative catchability. It would change if the abundance of other species in the catch changes, and as there are only 11 species considered, the catchability of each species is sensitive to how much of any other species is caught. Some of these species have declined substantially over the period, changing the estimated catchability of others. This issue with the method used should be discussed here. 

Lines 395-425: The ERA and Integrated ERA results here are in stark contrast to the IUCN ratings for several species: Carcharhinus longimanus (oceanic whitetip, OCS) and Sphyrna lewini (scalloped hammerhead, SPL) are critically endangered on the IUCN list, but both low on the ERA list, and the first is also in the group of least concern for the IERA list. Alopias pelagicus (pelagic thresher, PTH), C. obscurus (dusky shark, DUS) and Isurus oxyrinchus (shortfin mako shark, SMA) are in the lowest risk group in the ERA, yet all are recorded as endangered (C=0.8) on the IUCN list.

    The Authors discuss Sphyrna lewini but do not explain why their ERA would produce such a low risk even though the sizes of this species caught have declined dramatically over the period of the fishery considered. As suggested by my comment (lines 348-352) on the relative catchability parameter, is this perhaps because the depletion of the species has led to much lower catches in recent times, so that the relative catchability of this species has been under-estimated by their method? Selectivity, Post-capture mortality and Susceptibility are all relatively high (Table 5).

    The ERA for Carcharhinus falciformis is also a low value, although selectivity is high. The authors state the IERA ranking is high due to the decline in median weight over the fishing period. Could the low ERA also be due to an underestimate of the catchability due to depletion of this species?

    The authors also discuss their ERA rankings versus other assessments of Alopias superciliosus and Isurus oxyrinchus. But I suggest the disparity in assessment for Carcharhinus longimanusAlopias pelagicus, and C.obscurus, all should be explained. It is important to investigate whether the assessment used here might underestimate risks for these species.

Lines 427-432:  Management has to balance the collection of more data with costs. It would therefore be more useful to point out which kinds of data are most useful to gain a good assessment (or rsk analysis), than to say simply that biological and fisheries data should be updated.

The conclusion makes some very good points.

I have skimmed through some of the references and noted some errors, which suggest that a thorough check of the references is needed.

References 4 & 13: The journal Science is not usually abbreviated to “Sci”.

Refs 5 & 9: Similarly, Nature is not usually abbreviated.

Ref 7. “Fish” is an abbreviation of what? Fisheries? Or Fish and Fisheries?

Ref 15: ‘de Bruyn’ is the family name of the second author.

Ref 25 has the first names of authors rather than their last names.

Response: We have double checked the references and made necessary corrections and changes.